# Brief Exposure to Infants Activates Social and Intergroup Vigilance

**DOI:** 10.3390/bs10040072

**Published:** 2020-04-03

**Authors:** Bobby Cheon, Gianluca Esposito

**Affiliations:** 1School of Social Sciences (Psychology), Nanyang Technological University, Singapore 639798, Singapore; 2Singapore Institute for Clinical Sciences, Agency for Science Technology and Research (A*STAR), Singapore 138632, Singapore; 3Lee Kong Chian School of Medicine, Nanyang Technological University, Singapore 639798, Singapore; 4Department of Psychology and Cognitive Science, University of Trento, 38122 Trento, Italy

**Keywords:** parental care system, intergroup bias, infant exposure, social vigilance

## Abstract

Among humans, simply looking at infants can activate affiliative and nurturant behaviors. However, it remains unknown whether mere exposure to infants also activates other aspects of the caregiving motivational system, such as generalized defensiveness in the absence of immediate threats. Here, we demonstrate that simply viewing faces of infants (especially from the ingroup) may heighten vigilance against social threats and support for institutions that purportedly maintain security. Across two studies, participants viewed and rated one among several image types (between-subjects design): Infants, adult males, adult females, and puppies in Study 1, and infants of varying racial/ethnic groups (including one’s ingroup) and puppies in Study 2. Following exposure to one of these image types, participants completed measures of intergroup bias from a range of outgroups that differed in perceived threat, belief in a dangerous world, right-wing authoritarianism and social-political conservatism (relative to liberalism). In Study 1 (United States), stronger affiliative reactions to images of infants (but not adults or puppies) predicted stronger perceptions of a dangerous world, endorsement of right-wing authoritarianism, and support for social-political conservatism (relative to liberalism). Study 2 (Italy) revealed that exposure to images of ingroup infants (compared to outgroup infants) increased intergroup bias against outgroups that are characterized as threatening (immigrants and Arabs) and increased conservatism. These findings suggest a predisposed preparedness for social vigilance in the mere suggested presence of infants (e.g., viewing images) even in the absence of salient external threats.

## 1. Introduction

Among mammals, such as humans, survival and fitness of offspring are contingent on sufficient parental care. Parental care of offspring can be categorized into two broader classes of behavior: Nurturance and protection [1,2]. Nurturant care behaviors include activities such as feeding, grooming, and providing contact comfort, whereas protective care includes vigilant and defensive behaviors in anticipation or response to perceived threats. Likewise, offspring and infants are endowed with psychological and behavioral adaptations to maintain and bolster caregiving activities from caregivers through attachment behaviors [3,4], which along with the parental care system, may mutually reinforce sufficient care of offspring in response to situational demands.

While perceived needs or distress of offspring (hunger, exposure, fear, presence of harmful agents) are powerful triggers for activating the parental care system, it may have been especially adaptive for the system to spontaneously activate some degree of caregiving responses simply in the presence of offspring. Indeed, prior research has suggested that faces of infants have properties that may automatically activate a system of responses linked to caregiving from observers. Considering the evolutionary pathway of altricial human infant dependency on caregiving attitudes and behaviors, it is plausible that mechanisms associated with adult attentiveness and responsiveness to infants are embedded in caregivers’ nervous systems [5] and manifest even among non-parents [2]. Indeed, non-parent adults have specific implicit cognitive [6], autonomic [7,8], and central nervous system [9] reactions to human infant faces that differ from their responses to faces of human adults and faces of infants and adults of infrahuman mammals. Consequently, features of infants (i.e., faces) may act as powerful affordances that automatically activate and elicit caregiving behaviors. Yet, while the prepardness of activating nurturing caregiving tendencies in the presence of infants has been established, it remains critically unknown whether vigilant caregiving responses may also be activated by the mere presence of infants alone, even in the absence of salient perceived threats.

Activation of protective aspects of the parental care giving system based on mere exposure to infants would have been especially adaptive among mammals and ancestral humans. The central oxytonergic system, which is highly conserved across mammals and plays a critical role in regulating parental and social bonding tendencies, has been linked with maternal aggression towards intruders [10], suggesting that neurobiological mechanisms for parental care may also contribute to vigilance and defensive aggression. Hominids and early humans faced numerous dangers from other agents (animals and people), such as predation [11], intergroup conflict [12,13], and exploitation and harm from deviant or diseased group members [14]. Given that human infants are born relatively prematurely before sufficient brain or motor capacity has developed [15], they are especially vulnerable to potential harm from hostile agents. Consequently, the survival of infants may have required not only protectiveness of caregivers to immediate or salient threats, but also a chronic state of vigilance or defensiveness towards potential (rather than actual) threats in the presence of infants.

In the absence of an immediate threat that can be directly attacked, avoided, or removed, vigilance and protectiveness emerging from the presence of infants may manifest as relatively more generalized intentions to maintain safety from insecurity within the group and potentially hostile outsiders. This increased social vigilance may manifest in diverse ways—the first being general concerns or perceptions about the social world as being relatively more unstable and dangerous. Such worldviews may also fuel social ideologies, which represent systems of beliefs about how society should be properly ordered or structured [16,17]. In particular, concerns about threat and danger from the social world should increase support for patterns of social organization and institutions that maintain order, security, and suppress deviants and outsiders. Two such examples are political conservatism (relative to liberalism) and Right-Wing Authoritarianism (RWA), which reflect ideological systems with well-established functions of maintaining security, stability, and traditionally sanctioned conduct in response to perceived threats [17,18,19,20]. Finally, exposure to infants may also heighten perceptions of outgroup members (who historically represent intruders) as being sources of potential harm or exploitation. Notably, this effect should be more robust for outgroups that have been traditionally associated or stereotyped as being threatening, given prior research suggesting that heightened vigilance and sensitivity to threat may exclusively manifest as intergroup biases against outgroups perceived as formidable and dangerous [21,22,23,24].

Providing initial support for this proposal, prior work has suggested that situationally increasing salience of caregiving roles, such as by viewing or carrying infants, may increase social vigilance if such groups were also framed as a potentially immenent source of danger [25]. However, it remains unknown whether the mere perception of infants: (1) Produces biases against outgroups in the absence of reminders of any salient threats, (2) promotes more generalized ideological shifts that favor security and order, or (3) uniquely produces these outcomes compared to the perception of other targets that may also engender affiliative and caregiving motives (e.g., exposure to puppies).

Here, we tested whether mere exposure to images of infants activates increased vigilance towards potential social and intergroup threats. We predict that viewing faces of infants, particularly belonging to one’s ingroup, will subsequently heighten concerns about danger and threat from the social world, as well as increased support for policies or values that may promote social order and security. Specifically, we hypothesize that exposure to infants will lead to increased beliefs in a dangerous world (H1), greater endorsement of socio-politically conservative (H2) and authoritarian values (H3), and heightened intergroup bias against outgroup members (H4), especially those that have historically or culturally been stereotyped as dangerous. While we predict an *a priori* main effect of exposure to baby faces on these outcomes reflecting heightened social and intergroup vigilance, participants may experience differing levels of affiliative or caregiving motivations in response to infants, as well as variations in how readily the parental care motives are activated. Thus, we also conducted exploratory analyses to test whether our hypothesized relationships are moderated by the extent to which one experiences affiliative and approach-oriented responses to the infants one is exposed.

## 2. Study 1

In our first study (Study 1), participants from the United States completed an online study in which they were randomly assigned to view a group of faces from 4 categories: Infants, adult males, adult females, and puppies. Faces of puppies were included as a condition in the study to test whether exposure to human infants uniquely influences social/ideological attitudes, or whether these attitudinal shifts may more broadly emerge from exposure to vulnerable non-human targets that may also elicit tenderness and affiliative motives. Prior research has suggested that cute and vulnerable animals may also activate caregiving motives [26,27]. Yet, compared to parental care motives generated by human infants, these feelings produced by animals may be less likely to translate into types of social and intergroup vigilance examined in the present study. The threat of harm and exploitation from outgroup members may generally and automatically be expected to be directed at vulnerable human members of one’s ingroup (such as infants), rather than animals or pets. Furthermore, the types of security against social disorder and threats provided by conservative and authoritarian ideologies are largely exclusive for human beneficiaries rather than animals within a society (i.e., values and policies pertaining to abortion, immigration, enforcement of laws, upholding tradition, etc.). Therefore, although puppies may elicit feelings of warmth and tenderness, these feelings may be less likely to manifest and be applied to social and intergroup vigilance as compared to the similar experiences generated by infants.

After viewing and rating affiliative responses (warmth, closeness, desire to approach) for a series of face images that corresponded to one of the 4 categories, participants completed the outcome measures reflecting social and intergroup vigilance. These measures consisted of intergroup bias associated with various outgroups relative to their own ingroup, Belief in a Dangerous World (BDW) Scale, support for conservative relative to liberal policies and values, and the Right-Wing Authoritarianism Scale.

## 3. Methods

### 3.1. Participants

Participants included 404 adults (257 female, Mage = 39.30, SDage = 12.71) recruited from Amazon’s Mechanical Turk from the United States. A required total sample size of 280 participants was estimated based on our primary hypothesized analyses of a one-way ANOVA with four groups with a small to medium effect size of 0.20 and 80% power [28]. We recruited additional participants to reach a sample size of approximately 100 participants per condition to account for the actual effect size being weaker than our estimate of 0.20.The experiment was performed in accordance with regulations and guidelines regarding human subjects and was approved by the internal review board (IRB) of Nanyang Technological University (IRB-2015-08-025). Informed consent was obtained from all participants before they started the online survey.

### 3.2. Procedure

Participants were randomly assigned to one of four conditions in which they viewed a series of faces of White infants, White adult females, White adult males, or faces of puppies (10 faces in each condition, the stimuli were extracted from standardized faces dataset and used in previous studies) [7,9]. For each face, participants rated their feelings of coldness/warmth, closeness, and desire to avoid/approach the person on 100-point scales (higher values reflect more affiliative responses; Table 1). A composite index of mean affiliative motives towards the target of the images was computed by averaging participants’ ratings on these three dimensions across the faces they viewed (α = 0.91) (see Appendix A for results on this measure). Participants were free to view each face until they had completed the ratings associated with the face.

Participants then completed a feeling thermometer measure assessing their feelings of coldness/warmth towards various outgroups: Undocumented immigrants, Asian-Americans, White-Americans, African-Americans, Malaysians, Arabs, people from China, people with schizophrenia, and people who are the same ethnicity and nationality as the respondent (who represented the ingroup) (refer to Appendix A for results on these measures). These range of outgroup targets were selected given that they generally varied in stereotypical perceived threat and danger: Low (e.g., Asian-Americans), high (e.g., African-Americans, Arabs, undocumented immigrants, people with schizophrenia), and ambiguous (e.g., Malaysians). Following this, participants completed a measure of inter-group bias in the form of a series of semantic differential items in which they rated each target group on positive–negative traits (pleasant–unpleasant, nice–awful, safe–dangerous, moral–immoral, honest–dishonest). An index of inter-group bias for each target group was computed by a difference in average semantic differential ratings for the target group relative to one’s respective ingroup (higher values reflect stronger negativity towards the outgroup relative to one’s respective ingroup).

Participants also completed a measure of punitiveness towards people who commit violent crimes. Participants read two scenarios describing an incident when a male perpetrator physically assaulted a female victim, and was charged for the crime. For each scenario, the participant selected the appropriate sentence for the crime (ranging in severity from verbal reprimand to the death penalty). The perpetrator in the first scenario had a stereotypically White-American name, whereas the perpetrator in the second scenario had a stereotypically African-American name (refer to Appendix A for results on this measure).

Participants then completed a modified version of the 12-item Social Conservatism Scale [29], which involved rating agreement (on a 7-point Likert scale) to each policy or value, rather than a dichotomous ‘yes’/’no’ response (α = 0.85). Average level of agreement for conservative minus liberal values was computed as a measure of conservative (relative to liberal) ideology. Next, participants completed the Belief in a Dangerous World Scale (α = 0.94) [16], 15-item short version of the Right-Wing Authoritarianism Scale (α = 0.93) [30], Parental Bonding Instrument (α = 0.92) [31], and the Attachment Style Questionnaire (α = 0.83) [32] before completing general demographic questions (please see Appendix A for findings of the Parental Bonding Instrument and the Attachment Styles Questionnaire).

## 4. Results

We conducted a one-way ANOVA to test our *a priori* hypotheses of whether participants exhibited increased levels of BDW, conservatism, RWA, and perceived intergroup threat as a function of exposure to faces of infants (vs. males, females, or puppies). Overall, no significant differences in BDW, ideological variables, or perceived intergroup threat was observed as a function of the type of faces participants viewed (infants, females, males, or puppies), *p*’s > 0.10.

To test our exploratory predictions of whether our hypothesized relationships are moderated by the degree of affiliative responses experienced towards the faces, the PROCESS macro for SPSS [33] was used (model 1) to test the interaction between the affiliation composite and face condition (exposure to baby faces vs. non-baby faces) on BDW, RWA, conservatism, and perceived intergroup threat. In each model tested, the face condition (dummy coded to categorize exposure to infants or all other stimuli) was entered as the independent variable and the composite affiliation rating (corresponding to the faces participants viewed) was entered as the moderator to predict the outcome variables of BDW, RWA, conservatism, and intergroup bias (a separate model was run for each outcome variable).

The results revealed trending interactions of face condition and affiliative motives for BDW (model: F(3, 400) = 2.47, R2 = 0.02, *p* = 0.06; interaction: b = 0.019, *p* = 0.01) conservatism (model: F(3, 400) = 2.52, R2 = 0.02, *p* = 0.06; interaction: b = 0.014, *p* = 0.04), and a significant interaction for RWA (model: F(3, 400) = 3.85, R2 = 0.03, *p* = 0.01; interaction: b = 0.017, *p* = 0.02) (Figure 1). Affiliative motives in response to the faces predicted increased BDW, b = 0.015, *p* = 0.02, conservatism, b = 0.015, *p* = 0.009, and RWA, b = 0.02, *p* = 0.002 after exposure to faces of infants, but not after exposure to other types of faces (BDW: b = −0.003, *p* = 0.35; conservatism: b = 0.002, *p* = 0.64; RWA: b = 0.004, *p* = 0.33. There were no significant interactions between face condition and affiliative motives on intergroup bias against any outgroup, *p*’s > 0.10.

## 5. Discussion

Findings from Study 1 provide initial, albeit statistically weak, support for the hypothesis that affiliative feelings resulting from brief exposure to infants (compared to adult humans and puppies) increases perceptions of society as being more dangerous and unstable (H1), and support for ideologies and policies that promote security and suppress disorder, such as conservatism (H2) and RWA (H3). Conversely, we observed limited support for the hypothesis that exposure to infants heightens intergroup bias (H4).

Despite these trends that supported Hypotheses 1 to 3, we caution the reader that some of these relationships were not statistically robust. We conducted Study 2 in order to replicate these findings. Another objective of Study 2 was to examine whether the threat-sensitive ideological and attitudinal shifts resulting from exposure to infants is unique to ingroup infants, or infants more broadly (regardless of group membership). Furthermore, we also sought to test whether similar effects of ingroup infants on attitudes from Study 1 would also be observed in a different cultural and geopolitical landscape from the United States.

## 6. Study 2

Given historical human reliance on alloparenting and seeking support from other ingroup members for caring of offspring [34,35], defensive and vigilant responses following reminders of infants may largely occur following exposure to ingroup infants. Study 2 tested this prediction by briefly exposing participants to faces of infants that varied in racial ingroup/outgroup status (and puppies). Additionally, Study 2 involved a laboratory-based study that involved participants from a European (i.e., Italy) rather than North American context. We hypothesized that participants would exhibit the highest levels of belief in a dangerous world (H1), conservatism (H2), right-wing authoritarianism (H3), and intergroup bias (H4) from groups typically considered to be threatening after exposure to faces of ingroup infants relative to outgroup infants or puppies.

## 7. Methods

### 7.1. Participants

The final sample included 253 White participants recruited from an Italian university (males = 100, females = 153) (Mean age = 23.7, SD = 5.2). The experiment was performed in accordance with regulations and guidelines regarding human subjects, and being below minimal risks, application was waived (protocol of the Department of Psychology and Cognitive Sciences, University of Trento, Italy). Informed consent was obtained from all participants before they started the experimental session.

### 7.2. Procedure

The experiment consisted of an online survey involving self-reported questionnaires, which were presented in Italian language. Participants were randomly assigned to one of four conditions that involved viewing either White, Arab or Asian human infant faces, or puppies’ faces (eight images for each condition). For each face, participants made the same ratings of warmth, closeness, and approachability, which were also averaged into a composite index of affiliative responses (α = 0.93; Table 2) as in Study 1.

Participants then completed the same feeling thermometer and semantic differential measures from Study 1 for the following target groups: Whites, Africans, Asians, undocumented immigrants, Chinese, Malaysians, Arabs, people with schizophrenia, and people who are the same ethnicity and nationality as the respondent (who represented the respective ingroup). An index of intergroup bias toward each target group was computed in the same manner as Study 1, in which semantic differential items that assess associations of positive–negative traits (pleasant–unpleasant, nice–awful, safe–dangerous, moral–immoral, honest–dishonest) were averaged for each target group and subtracted from the average ratings for one’s respective ingroup (higher values reflect more negative biases against outgroups). Next, participants completed the same measures of conservatism/liberalism and belief in a dangerous world (α = 0.79) as in Study 1. Participants also completed the Competitive Jungle Beliefs Questionnaire (α = 0.86) [16] (see Appendix A for results on this measure). This was followed by the Italian version of the short 12-item version of the RWA scale (α = 0.63) [36,37]. Finally, as in Study 1, participants completed the Parental Bonding Instrument (α = 0.92) and Attachment Style Questionnaire (α = 0.77) before the general demographic questions (please see Appendix A for findings of the Parental Bonding Instrument and the Attachment Styles Questionnaire).

## 8. Results

We conducted a one-way ANOVA to test whether participants exhibited increased levels of BDW, competitive jungle beliefs, conservatism, RWA, and intergroup bias as a function of exposure to faces of white infants (vs. Asian infants, Arab infants, or puppies).

A one-way ANOVA revealed marginally significant trends suggesting that participants attributed different levels of intergroup bias (based on semantic differential ratings) to undocumented immigrants, F(3, 249) = 2.23, *p* = 0.09, *d* = 0.03, and Arabs, F(3, 249) = 2.63, *p* = 0.05, *d* = 0.03, as a function of condition (Figure 2A,B). Paired comparisons showed that participants viewing White infants (M = 1.53, SD = 1.56) attributed greater levels of intergroup bias towards undocumented immigrants than participants who viewed Arab infants (M = 0.90, SD = 1.48), *p* = 0.02, *d* = 0.41, Asian infants (M = 1.01, SD = 1.35), *p* = 0.046, *d* = 0.36, but not puppies (M = 1.11, SD = 1.46), *p* = 0.12, *d* = 0.28. Likewise, participants exposed to White infants (M = 1.44, SD = 1.42) also exhibited greater biases against Arabs compared to participants who viewed Arab infants (M = 0.81, SD = 1.53), *p* = 0.009, *d* = 0.43, Asian infants (M = 0.92, M = 1.12), *p* = 0.03, *d* = 0.41, but not puppies (M = 1.10, SD = 1.39), *p* = 0.17, *d* = 0.24. Notably, from all the target groups that were assessed in the intergroup bias measures, undocumented immigrants and Arabs were overall associated with the most negative ratings on the semantic differentials, F(8, 2016) = 61.80, *p* < 0.001, *d* = 0.20, such that these two groups were attributed significantly higher levels of negative/threatening traits relative to all other groups, *p*’s < 0.005, but they did not significantly differ from each other in ratings, t(252) = 1.34, *p* = 0.18, *d* = 0.17. There were no significant effects of condition on intergroup bias towards any other targets, *p*’s > 0.10.

Participants also exhibited a marginally significant trend of endorsing differing levels of conservative (relative to liberal) socio-political attitudes based on condition, F(3, 249) = 2.61, *p* = 0.052, *d* = 0.03. Paired comparisons revealed that participants viewing White infants (M = −1.08, SD = 0.96) endorsed greater levels of conservatism (vs. liberalism) relative to participants who viewed Arab infants (M = −1.40, SD = 0.70), *p* = 0.03, *d* = 0.38, Asian infants (M = −1.33, SD = 0.96), *p* = 0.09, *d* = 0.26, and puppies (M = −1.47, SD = 0.76), *p* = 0.01, *d* = 0.45 (Figure 2C). Participants did not differ in endorsement of BDW, or RWA as a function of condition, *p*’s > 0.10.

As in Study 1, the PROCESS macro for SPSS [33] was used (model 1) to test the interaction between the affiliation composite and face condition (exposure to white infant faces vs. all other face types) on BDW, competitive jungle beliefs, RWA, conservatism, and perceived intergroup threat. Unlike Study 1, we observed no overall significant interactions on intergroup bias or social attitude measures, *p*’s > 0.10.

## 9. Discussion

Overall, Study 2 provides further support for our hypotheses although it did not yield a direct conceptual replication of Study 1’s findings. Unlike Study 1, we observed the primarily hypothesized main effects of exposure to ingroup infant faces on endorsement of conservative (relative to liberal) values (Hypothesis 1) and greater intergroup bias selectively against outgroups stereotyped as threatening (Arabs and undocumented immigrants) (Hypothesis 4). This inconsistency in the pattern of results between Studies 1 and 2 may be attributed to the differences between the studies in sample characteristics and societal contexts (e.g., American participants recruited for an online survey versus Italian university students). Preliminary support for Hypothesis 4 in Study 2, which was not observed in Study 1, also suggests that brief exposure to infants may indeed activate social vigilance that may also manifest as intergroup bias directed against potentially threatening groups.

## 10. General Discussion and Conclusions

Across two studies conducted in different societies, we find support for our hypotheses that brief exposure or reminders (i.e., faces) of ingroup infants activates generalized social and intergroup vigilance. Depending on the culture or societal context, this heightened vigilance may manifest as increased perceptions of danger and disorder from the social environment (Study 1), ideological shifts towards conservatism (relative to liberalism) (Studies 1 and 2), increased support of right-wing authoritarianism (Study 1), and heighted intergroup bias directed at outgroups typically stereotyped as threatening (e.g., undocumented immigrants and Arabs in a European context) (Study 2). We caution the reader that some of these findings were not statistically robust. Together, these findings provide preliminary, yet novel and promising, insights that suggest perceptual properties of infants may not only spontaneously activate affiliative or nurturing responses, but also generalized vigilance that is directed at potential sources of threat from the social world, which may have downstream influences on socio-political ideologies.

Conventionally, ideological and attitudinal shifts towards social and intergroup vigilance have typically been linked with perceptions of threat and insecurity [18,19,38,39]. However, our findings reveal that in the absence of any salient sources of external threat and insecurity, perceptual features of infants (especially from the ingroup) are also capable of producing such defensive shifts in ideological and intergroup attitudes. Infants historically represent exceptionally vulnerable members of the group that elicit feelings of warmth, affiliation, and concern. Importantly, the finding that outgroup infants and puppies (who overall elicited the highest levels of affiliative reactions) did not generate the same shifts towards ideological and intergroup vigilance reveals that up-regulation of generalized affiliative motivations are alone insufficient to produce these attitudinal shifts. Rather, such affiliative states may need to be specifically activated by or directed towards vulnerable members of the ingroup (infants) for this vigilance to emerge. This is consistent with prior research suggesting that the affiliation- and cooperation-inducing properties of oxytocin are especially selective for ingroup targets [40,41]. Moreover, although humans have historically provided altruistic caregiving of unrelated offspring [34,35], this tendency for alloparenting may have been exclusively restricted to offspring of other ingroup members. Given that these responses to ingroup infants were observed even among non-parents (Study 2) and across both genders, these findings suggest an adaptive preparedness or predisposition of human adults to activate defensive/protective properties of the parental care system in the presence of infants identified with ingroup membership.

While we observed overall convergent support for our hypotheses across the two studies, some patterns of results were less consistent across the two samples and cultures. In Study 1, we observed interactions between overall affiliative reactions towards the faces and the presence (vs. absence) of infant faces on outcomes of interest, whereas in Study 2, the mere presence of ingroup infant faces (regardless of affiliative reactions) were sufficient to produce shifts in outcomes. Furthermore, the outcomes affected by exposure to infant faces differed across the two studies, such that Study 1 revealed greater shifts in perceptions and ideology linked to broader social environment and institutions (i.e., belief in dangerous world, political ideology, right-wing authoritarianism), whereas Study 2 revealed greater shifts in perceptions of intergroup threat. While these two studies still convey overall the effect of exposure to infants on vigiliance towards social threats, their discrepancies may be a reflection of the different sociopolitical climates of the United States (Study 1) and Italy (Study 2), and variations in the social/ideological concerns prioritized across the two societies. Furthermore, the two studies also varied in design and sample characteristics, such that Study 1 recruited a demographically broader sample while Study 2 focused on a predominantly younger student sample, which may have produced differences in the types of social threats (ideological vs. intergroup) most salient to participants between studies. Finally, some of these findings were not highly robust. This could be a product of the brief and subtle manipulation used in the studies (simply viewing and rating images of infants) and the absence of other salient reminders of threat. This is in contrast to other prior studies that involved more heavy-handed and salient reminders of caregiving roles (e.g., writing about being a parent, being in the presence of one’s child) or included a direct manipulation of salient outgroup threat (see [25]). Yet due to the design of our current studies, we were able to establish initial support that mere reminders of infants (particularly of the ingroup) is sufficient to heighten social and intergroup vigilance even under such neutral and minimal conditions.

These initial findings prompt further avenues for investigation. One important question is whether these effects are exclusive to faces of infants, or whether they can be activated by other perceptual cues. For instance, vocalizations (especially crying) of infants are a powerful stimulus [5], and may also produce similar defensive motives and mindsets. Another promising avenue for future research is identifying the age-related boundary conditions in which mere exposure to an infant or child no longer generates these spontaneous vigilant responses from perceivers.

Prior studies examining the relationship between becoming a first-time parent and sociopolitical ideology have yielded inconsistent results [42]. Despite this, our findings suggest that shifts in social, political, and intergroup perceptions towards vigilance and maintenance of security may occur even in response to minimal exposure to others’ infants. In conclusion, we demonstrate how even such complex ideological systems and worldviews may be served by and rooted in fundamental motives for parental caregiving.

## Figures and Tables

**Figure 1 behavsci-10-00072-f001:**
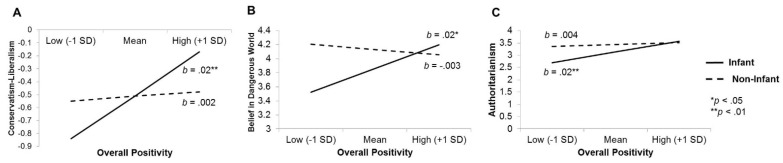
Interactions between the type of face participants viewed (infants or non-infants) and overall positivity (affiliative reactions) to the faces on worldviews and ideologies in Study 1. Among participants who were exposed to faces of infants, higher overall affiliative reactions were associated with increased: (**A**) Support for conservative (relative to liberal) values, (**B**) perceptions of a dangerous world, and (**C**) endorsement of right-wing authoritarianism. Note: Values for conservatism (relative to liberalism) are negative given that the sample overall endorsed stronger liberal values than conservative ones.

**Figure 2 behavsci-10-00072-f002:**
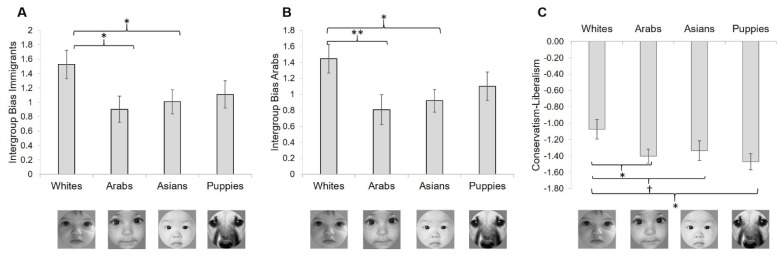
Effects of the type of face viewed on: (**A**) Perceived threat from undocumented immigrants (relative to one’s ingroup), (**B**) perceived threat from Arabs (relative to one’s ingroup), and (**C**) support for conservative (relative to liberal) values. Note: Values for conservatism (relative to liberalism) are negative given that the sample overall endorsed stronger liberal values than conservative ones († *p* < 0.10, * *p* < 0.05, ** *p* < 0.01).

**Table 1 behavsci-10-00072-t001:** Means and standard deviations (in parentheses) across the four face stimuli conditions (females, males, infants, and puppies). Asterisk notation (*) indicates that the mean for a given facial stimuli condition (females, males, puppies) differs significantly from the infant facial stimuli condition (*p* < 0.05).

	Infants	Males	Females	Puppies
Warmth	76.00 (2.12)	51.21 (11.21) *	55.02 (13.83) *	84.74 (1.46) *
Closeness	55.91 (27.65)	36.92 (20.89) *	40.26(21.21) *	84.74 (1.46) *
Desire for Approach	67.09 (25.42)	49.17 (14.97) *	53.98 (14.14) *	79.20 (22.68) *
Overall Positivity	66.33 (21.90)	45.77 (13.65) *	49.75 (14.54) *	78.40 (19.72) *
Conservatism-Liberalism	−0.42 (1.22)	−0.46 (1.19)	−0.58 (1.41)	−0.50 (1.16)
Belief in Dangerous World	3.95 (1.42)	4.19 (1.37)	4.22 (1.50)	4.01 (1.44)
Right-Wing Authoritarianism	3.32 (1.34)	3.40 (1.24)	3.42 (1.51)	3.47 (1.34)

**Table 2 behavsci-10-00072-t002:** Means and standard deviations (in parentheses) across the four face stimuli conditions (White infants, Arab infants, Asian infants, and puppies). Asterisk notation (*) indicates that the mean for a given facial stimuli condition (Arab infants, Asian infants, puppies) differs significantly from the White infant facial stimuli condition (*p* < 0.05).

	Whites	Arabs	Asians	Puppies
Warmth	54.28 (12.58)	51.91 (13.96)	56.76 (17.35)	61.43 (20.41) *
Closeness	59.15 (15.52)	58.07 (18.70)	57.90 (20.65)	69.95 (21.87) *
Desire for Approach	58.61 (15.02)	56.05 (16.48)	59.60 (21.37)	60.68 (18.71) *
Overall Positivity	57.35 (13.54)	55.34 (15.51)	57.93 (18.15)	66.99 (19.48) *
Conservatism-Liberalism	−1.07 (.96)	−1.40 (0.7) *	−1.33 (0.96)	−1.47 (0.76) *
Belief in Dangerous World	3.23 (0.73)	3.24 (0.57)	3.38 (0.63)	3.41 (0.68)
Right-Wing Authoritarianism	3.12 (0.80)	3.16 (0.76)	3.20 (0.73)	3.07 (0.84)
White-Intergroup Bias	0.57 (1.05)	0.30 (0.63)	0.33 (0.68)	0.40 (0.99)
Black-Intergroup Bias	0.73 (1.18)	0.40 (1.16)	0.44 (0.97)	0.64 (1.41)
Asian-Intergroup Bias	0.83 (1.00)	0.46 (1.07)	0.50 (0.85)	0.63 (1.18)
Immigrant-Intergroup Bias	1.53 (1.56)	0.90 (1.48) *	1.01 (1.35) *	1.11 (1.46)
Malay-Intergroup Bias	0.69 (0.96)	0.41 (0.93)	0.44 (0.97)	0.59 (1.13)
Arab-Intergroup Bias	1.44 (1.42)	0.81 (1.53) *	0.92 (1.12) *	1.10 (1.39)
Schizophrenia-Intergroup Bias	1.00 (1.19)	0.85 (1.09)	0.82 (1.22)	0.76 (1.14)
China-Intergroup Bias	0.72 (1.12)	0.46 (1.04)	0.44 (0.88)	0.50 (0.97)

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
