# Peer review of "Brief Exposure to Infants Activates Social and Intergroup Vigilance"

_behavsci, 2020, doi:10.3390/bs10040072_

Round 1

Reviewer 1 Report

Thank you for giving me the opportunity to review the manuscript entitled “Brief Exposure to Infants Activates Social and Intergroup Vigilance”

Please find below several comments which in my opinion can improve the paper:

  • Please consider the modification of the title to make it more clear. What does it mean “Brief Exposure to Infants”?
  • Based on the title and even abstract it is not clear if the current study is the original research or the review of existing knowledge
  • The abstract need to be rewritten, The aim of the study needs to be clearly stated. What is more I could not find the methodology of the study. So the abstract (even without subheadings) should contain the following parts: background (with clear aim of the study), methods, results and conclusions.
  • The methods, results and discussion parts are not clear – they need to be improved. Why do not include Study 1 and Study 2 under the methods, results and discussion? So do not describe study 1 and study 2 separately. I propose to write the Methods part in which in the first sentence it is stated that 2 studies were conducted and then each of them are described. The same for the results and discussion – under the one subheading (Results and then Discussion) two studies were described.
  • The separate subheading “statistical analysis” needs to be added under the methods.
  • I would also add one discussion for the whole paper
  • Final section “conclusions” need to be added.

Reviewer 2 Report

In principle, the topic of this research manuscript is interesting and the manuscript itself is well written.

My main concern regards the two different cohort of subjects enrolled. They are too much dissimilar. Not only in America and Europe (Italy). Age also is too different, as social status.

In addition, did the subjects have sons? This could be another bias.

Author Response

In principle, the topic of this research manuscript is interesting and the manuscript itself is well written.

My main concern regards the two different cohort of subjects enrolled. They are too much dissimilar. Not only in America and Europe (Italy). Age also is too different, as social status.

We agree that there are some systematic differences between the Singaporean and Italian cohorts. As such, the two cohorts are NOT being presented as a direct cross-cultural comparison. Thus, they are presented as two separate studies (study 1 and 2) on a similar psychological process. The two samples are also supposed to have different characteristics, since this allows us to test our research question under new conditions in Study 2. Most notably, study 2 involves a different design as well, such that participants viewed infants of different ethnic/racial groups rather than infants versus other targets (e.g., adults). Furthermore, we intentionally selected a younger student sample in Study 2 because we sought to largely isolate and focus on participants who would not have children. This allowed us to demonstrate that one does not need to be a parent to show changes in social attitudes in response to infants.

Our discussion section (page 9) mentions these differences between the sample characteristics of the 2 studies, and how some of the inconsistent patterns of results in the two studies could be due to these differences in the cohorts.

In addition, did the subjects have sons? This could be another bias.

We believe that the gender of the participants’ children, or whether they even have children of their own, may not have a major biasing effect on the results. As indicated above, one of the reasons that we selectively recruited a young adult sample of university-age in Study 2 (rather than a broader sample of adult age ranges) was to test whether the psychological processes we are studying are observed even in non-parents. As noted in our introduction to Study 2 and in the general discussion, dependence on alloparenting among humans required people to engage caregiving motivations and actions for children who were not their own. Thus, people who do not have children of their own may still exhibit a tendency to activate caregiving motivations (such as defensiveness and vigilance) when exposed to representations or reminders of infants.

Round 2

Reviewer 1 Report

The authors have not even tried to follow my recommendations so I am not able to make the decision of the acceptance of this paper.

Author Response

Dear Reviewer

we followed all your requests before. I had attached a file with our answers. I am typing here our replies. I apologize for the misinderstanding

Thank you

Please consider the modification of the title to make it more clear. What does it mean “Brief Exposure to Infants”?

We use the terms “brief exposure to infants” to refer to the design of our study. Here, participants have only superficial exposure to infants in the form of photographs they view and rate. This brief and minimal exposure to representations of infants is sufficient to produce the shifts in social attitudes we observe in our study. The terms “brief exposure” contrasts our design from other thematically similar studies that are described in the introduction, such as prior studies that show changes in social attitudes after much more substantive, salient and prolonged caregiving situations such as carrying one’s own infant or being the direct presence of infants (Gilead & Liberman, 2014). We have updated the last sentence of our abstract to make it clearer what we mean by brief exposure (e.g., mere suggestions that there are infants present, such as exposure to images).

Based on the title and even abstract it is not clear if the current study is the original research or the review of existing knowledge

The abstract makes it clear that this is original research in that it specifically describes the purpose or intent of an empirical demonstration that exposure to infants influences social attitudes. Furthermore, the abstract specifically states the results of the two studies that were conducted in this manuscript (with labels of Study 1 and Study 2).

The abstract need to be rewritten, The aim of the study needs to be clearly stated. What is more I could not find the methodology of the study. So the abstract (even without subheadings) should contain the following parts: background (with clear aim of the study), methods, results and conclusions.

We appreciate the reviewer’s suggestions for further clarify on the abstract. Although some journals apply the format of having subheadings within their abstracts (i.e., background, objectives, methods, results, etc..), the present journal, Behavioral Sciences, does not. Examining many recent articles that have been published (during March 2020) reveals that none of these recent articles have used the format of subheadings within the abstract. Thus to ensure consistency with the general format of abstracts in recent articles, we have kept the abstract without subheadings.

That said, we have added some sentences in the abstract to reflect the methodology of the study. The revised abstract is below (with new sentences highlighted):

Among humans, simply looking at infants can activate affiliative and nurturant behaviors. Yet it remains unknown whether mere exposure to infants also activates other aspects of the caregiving motivational system, such as generalized defensiveness in the absence of immediate threats. Here, we demonstrate that simply viewing faces of infants (especially from the ingroup) may heighten vigilance against social threats and support for institutions that purportedly maintain security. Across two studies, participants viewed and rated one among several image types (between-subjects design): infants, adult males, adult females, and puppies in Study 1, and infants of varying racial/ethnic groups (including one’s ingroup) and puppies in Study 2. Following exposure to one of these image types, participants completed measures of intergroup bias from a range of outgroups that differed in perceived threat, belief in a dangerous world, right-wing authoritarianism and social-political conservatism (relative to liberalism). Study 1 (United States), stronger affiliative reactions to images of infants (but not adults or puppies) predicted stronger perceptions of a dangerous world, endorsement of right-wing authoritarianism, and support for social-political conservatism (relative to liberalism). Study 2 (Italy) revealed that exposure to images of ingroup infants (compared to outgroup infants) increased intergroup bias against outgroups characterized as threatening (immigrants and Arabs) and increased conservatism. These findings suggest a predisposed preparedness for social vigilance in response to the mere suggested presence of infants (e.g., viewing images), even in the absence of salient external threats.

The methods, results and discussion parts are not clear – they need to be improved. Why do not include Study 1 and Study 2 under the methods, results and discussion? So do not describe study 1 and study 2 separately. I propose to write the Methods part in which in the first sentence it is stated that 2 studies were conducted and then each of them are described. The same for the results and discussion – under the one subheading (Results and then Discussion) two studies were described.

We thank the reviewer for this suggestion. Similar to our response about formatting of the abstract with subheadings, the present journal does not seem to mandate multi-study manuscripts to be structured so that methods, results and discussion for separate studies are described simultaneously rather than sequentially. Typically, for journals in psychology (particularly social psychology), multi-study papers are structured so that there is a common introduction, methods/results/discussion are presented separately and sequentially for each study, and there is a common general discussion. Given that the present manuscript could be primarily classified as social psychological, we expect most readers of this article to also be accustomed to this style of formatting. Thus, we left the overall structure of the manuscript unchanged.

The separate subheading “statistical analysis” needs to be added under the methods.

As the journal does not consider this subheading necessary we did not include it.

I would also add one discussion for the whole paper

The manuscript indeed has one discussion that covers the whole paper (‘general discussion’) on page 8. This is in addition to shorter, more study-specific discussions that follow each of our two studies.

Final section “conclusions” need to be added.

The general discussion also serves as a conclusions section to the manuscript. We have relabeled it as “General Discussion and Conclusions.”

Reviewer 2 Report

Authors well answered to my comments.

Author Response

Thank you